# Anti-inflammatory activity of the dietary supplement *Houttuynia cordata* fermentation product in RAW264.7 cells and Wistar rats

Khanutsanan Woranam[1], Gulsiri Senawong[1], Suppawit Utaiwat[1], Sirinda Yunchalard[2], Jintana Sattayasai[3], Thanaset Senawong[1,4]*

1 Department of Biochemistry, Faculty of Science, Khon Kaen University, Khon Kaen, Thailand,
2 Department of Biotechnology, Faculty of Technology, Khon Kaen University, Khon Kaen, Thailand,
3 Department of Pharmacology, Faculty of Medicine, Khon Kaen University, Khon Kaen, Thailand, 4 Natural Product Research Unit, Faculty of Science, Khon Kaen University, Khon Kaen, Thailand

* sthanaset@kku.ac.th

## Abstract

*Houttuynia cordata* Thunb. has been used as a traditional medicine to treat a variety of ailments in Asian countries such as China, Japan, South Korea, and Thailand. In Thailand, *H. cordata* fermentation products (HCFPs) are commercially produced and popularly consumed throughout the country without experimental validation. Anti-inflammatory activity of *H. cordata* fresh leaves or aerial parts has previously been reported, however, the anti-inflammatory activity of the commercially available HCFPs produced by the industrialized process has not yet been investigated. The aim of this study was to evaluate *in vitro* and *in vivo* anti-inflammatory potential of the selected industrialized HCFP. LPS-induced RAW264.7 and carrageenan-induced paw edema models were used to evaluate the anti-inflammatory activity of HCFP. The phenolic acid components of HCFP aqueous and methanolic extracts were investigated using HPLC analysis. In RAW264.7 cells, the HCFP aqueous and methanolic extracts reduced NO production and suppressed LPS-stimulated expression of $PGE_2$, iNOS, IL-1β, TNF-α and IL-6 levels in a concentration-dependent manner, however, less effect on COX-2 level was observed. In Wistar rats, 3.08 and 6.16 mL/kg HCFP reduced paw edema after 2 h carrageenan stimulation, suggesting the second phase anti-edematous effect similar to diclofenac (150 mg/kg). Whereas, 6.16 mL/kg HCFP also reduced paw edema after 1 h carrageenan stimulation, suggesting the first phase anti-edematous effect. Quantitative HPLC revealed the active phenolic compounds including syringic, vanillic, *p*-hydroxybenzoic and ferulic acids, which possess anti-inflammatory activity. Our results demonstrated for the first time the anti-inflammatory activity of the industrialized HCFP both *in vitro* and *in vivo*, thus validating its promising anti-inflammation potential.

## Introduction

Inflammation is a host response against infection, foreign stimulant and tissue injury. Although inflammation is a process of the immune response in our body, it can damage the body when it is out of control. While acute inflammation is a normal part of the defense

**Data Availability Statement:** All relevant data are within the manuscript and its Supporting Information files.

**Funding:** This study was supported by the Research and Researcher for Industry (RRi) project, Thailand Research Fund (TRF), which was cooperated with the Prolac (Thailand) Co., Ltd., Lamphun Province, Thailand. The funders had no role in study design, data collection and analysis, decision to publish, or preparation of the manuscript. However, the funders have been informed and agreed on publishing a manuscript, sharing data and materials.

**Competing interests:** I would like to undertake that all of authors have no relation with the Prolac (Thailand) Co., Ltd., who is a co-funder with Thailand Research Fund (TRF) on the Research and Researcher for Industry (RRi) project. The company provided 60,000 baht in cash and the HCFP samples (product Lot no. 14/5/2015), while the TRF provided a Ph.D. scholarship and major funding source for the project. This does not alter our adherence to PLOS ONE policies on sharing data and materials.

**Abbreviations:** COX-2, Cyclooxygenase-2; DMSO, Dimethyl sulfoxide; HCFP, H. cordata fermentation product; iNOS, Inducible nitric oxide synthase; IL, Interleukin; LPS, Lipopolysaccharide; MTT, 3-(4,5-dimethylthiazol-2-yl)-2,5-diphenyltetrazolium bromide; NO, Nitric oxide; PGE$_2$, Prostaglandin E2; PVDF, Polyvinylidene fluoride; TNF, Tumor necrosis factor.

response, chronic inflammation is a complex process stimulated by activating inflammation or immune cells. During the inflammatory process, many types of cells are activated, and these cells secrete various pro-inflammatory mediators, including cytokines (IL-1β, TNF-α, IL-6), nitric oxide (NO) and prostaglandin E2 (PGE$_2$) [1]. Overproduction of inflammatory mediators leads to chronic inflammation, which can cause many diseases such as rheumatoid arthritis, cancer and allergies [2]. During the inflammatory response, immune cells are also activated by adhesion molecules-activated signals to increase the migration capacity to inflamed tissue and finally to form heterotypic cell clustering between the immune cells, endothelial cells, and inflamed cells. Indeed, various inflammatory stimuli such as LPS and pro-inflammatory cytokines activate immune cells to up-regulate such inflammatory states [2,3]. Hence, these cells are useful targets for developing new anti-inflammatory drugs and exploring the molecular anti-inflammatory mechanisms of a potential drug.

Many drugs have been developed to treat inflammation and nociceptive symptoms however, undesired adverse effects of these clinical anti-inflammatory drugs have consistently evidenced [4]. Non-steroidal anti-inflammatory drugs (NSAIDs) are normally used for the treatment of pain and inflammatory conditions, however, many NSAIDs are associated with undesired side effects including congestive heart failure, bleeding of the gastrointestinal tract and chronic kidney disease [5]. Therefore, the search for alternative substitutes such as plant-derived anti-inflammatory agents with the ease of availability and fewer side effects is urgently required to develop safe drugs for clinical use.

*Houttuynia cordata* Thunb. is a perennial herbaceous plant mostly distributed in East Asia, and generally grown for local vegetable consumption in the North and Northeast of Thailand. *H. cordata* has been used as a medicinal plant possessing many biological properties including antioxidant, anticancer, and anti-inflammatory activities [6]. As a traditional medicine in China, *H. cordata* has been used to treat ulcers, sores, heatstroke, diarrhea, and dysentery [7]. In Korea, it has been used for the treatment of pneumonia, bronchitis, dysentery, dropsy, uteritis, eczema, herpes simplex, chronic sinusitis and nasal polyps [8]. In Thailand, it has been used as an immunostimulant herb and anticancer agent [9].

Nowadays, *H. cordata* is considered for a high-value industrial crop in Thailand and it has been fermented with probiotic bacteria to yield a *H. cordata* fermentation product (HCFP) commercially available. The microbial fermented herbal plant is a promising alternative source for many flavonoid molecules including anthocyanins, flavones and flavanones [10]. Probiotics are microorganism exerting health-promoting functions in humans and animals [11], improving the nutraceutical value of the herbal plant products by breaking down undesirable phytochemicals, and producing certain desirable compounds [12]. The fermentation process has increased the flavonoid content of *H. cordata* fermentation products conferring excellent anti-inflammatory effects in LPS-stimulated cells [13]. Accordingly, many HCFPs have been commercially distributed and popularly consumed throughout Thailand. Previous studies reported that the industrial process caused a reduction in phenolic content of natural products [14,15], which may affect their biological properties. Anti-inflammatory activity of *H. cordata* fresh leaves or aerial parts has previously been studied [16,17,18], however, the anti-inflammatory activity of HCFPs produced by industrialized process has not yet been investigated. Therefore, we aimed at investigating the anti-inflammatory activity of the industrial HCFP in LPS-stimulated RAW264.7 cells as well as its phenolic acid content to provide information for the general public or consumers. Here, we demonstrated the phenolic acid profiles and anti-inflammatory activities of aqueous and methanolic extracts of the industrial HCFP Dokudami manifested by inhibiting the production of NO, PGE$_2$ and inflammatory cytokines such as TNF-α, IL-1β, and IL-6. Furthermore, the anti-inflammatory activity of this industrialized product was also confirmed using the rat paw assay.

## Materials and methods

### Materials

The dietary supplement *H. cordata* fermentation product (HCFP), Dokudami, was obtained from the Prolac (Thailand) Co., Ltd., Lamphun, Thailand. The information on plant ingredient and serving suggestion of the HCFP were obtained from the label on its container. The major ingredients of this HCFP are composed of 99.3% (w/w) aerial parts of *H. cordata* and 0.7% (w/w) sugar cane powder. Serving suggestion is as follows: 5–15 ml twice a day in the morning before bedtime and before meal. *H. cordata* was cultivated by the Prolac (Thailand) Co., Ltd. in an organic farm in Chai Badan district, Lopburi province, Thailand. The fermentation product Lot no. 14/5/2015 was used throughout the study. RAW264.7 cells were obtained from Dr. Pramote Mahakunakorn, Faculty of Pharmaceutical Science, Khon Kaen University, Thailand. Male Wistar rats (250–300 g) were obtained from registered animal breeders, Nomura Siam International Co., Ltd., Bangkok, Thailand. LPS (*E. coli* 0111: B4) and diclofenac sodium were purchased from Sigma-Aldrich (St. Louis, MO, USA). Griess reagent for nitrite determination was purchased from Molecular Probes (Invitrogen, USA). All antibodies used in this study were purchased from Cell Signaling (USA). $PGE_2$ EIA was purchased from ANOVA (Taiwan). The ELISA kits for measuring cytokines (IL-1β, TNF-α, IL-6) were purchased from BioLegend (California). RPMI 1640 medium, fetal bovine serum (FBS), trypsin-EDTA and penicillin/streptomycin were obtained from Gibco/Invitrogen Crop. (Grand Island, NY, USA).

### Cell culture and animals

RAW264.7 macrophage cells were cultured in RPMI 1640 medium with 10% fetal bovine serum (FBS), 1% penicillin and streptomycin and incubated at 37˚C in an atmosphere containing 5% $CO_2$. Wistar rats were recovered from transportation for 1 week before the study. The rats were maintained at Northeast Laboratory Center, Khon Kaen University, Thailand. Details of animal welfare are as follows: Shelter: case size 37.5 x 48 x 18.5 cm (wide x length x high), with sterilized-wood shavings for bedding, Food: sterilized commercial food, Water: reverse osmosis (OR) with choline 3–4 ppm, Environment enrichment: social housing, free excess food, and water, Environment: temperature: 23±2˚C, humidity: 30–60% RH, dark: light cycle: 12:12 h, illumination: 350–400 Lux, ventilation: 10–15 ACH, noise: no exceed 85 Decibels. The experimental procedure was approved by the Institutional Animal Care and Use Committee of Khon Kaen University, based on the Ethic of Animal Experimentation of National Research Council of Thailand. The approval number was IACUC-KKU-101/60.

### Preparation of the lyophilized powder of HCFP aqueous extract

To obtain polar phytochemical compounds, 50 mL of HCFP was centrifuged at 2,815 x *g* for 15 min and the supernatant (aqueous fraction) was filtered through Whatman grade No. 4 filter paper. The filtrate containing water-soluble constituents was lyophilized to obtain a lyophilized powder (aqueous extract). The extraction yield was 13.80 ± 0.57 mg/mL. The HCFP lyophilized powder was re-dissolved in double distilled water to obtain desired concentrations.

### Preparation of HCFP methanolic (phenolic-rich) extract

To obtain both polar and nonpolar phenolic compounds in the free forms, 140 mL of methanol was added to 60 mL of HCFP and then the mixture was stirred for 2 h at room temperature. The filtrate was evaporated to 60 mL by rotary evaporator, then added with 60 mL of 2 M NaOH and stirred continuously for 12 h at room temperature. The mixture was centrifuged at

1,700 x $g$ for 20 min and then filtered through Whatman grade No. 4 filter paper. The supernatant was repeatedly extracted three times with 80 mL of diethyl ether and the aqueous phase was collected and the diethyl ether phase was discarded. The aqueous phase was adjusted to pH 1.5 by 10 M HCl and filtered through Whatman grade No. 4 filter paper. The filtrate was extracted further with 80 mL of diethyl ether for three times, in which the portion of diethyl ether was collected. Sodium sulphate ($Na_2SO_4$) anhydrous was used to dehydrate the diethyl ether phase, which was then filtered through the filter paper. A rotary evaporator was used to evaporate the filtrate to 5 mL, which was then finally evaporated to dryness under a gentle stream of nitrogen gas. The extraction yield was 5.36 ± 0.96 mg/mL.

## Cell viability assay

The viability of RAW264.7 cells was determined colorimetrically using 3-(4,5-dimethylthiazo-lyl)-2-2,5-diphenyltetrazolium bromide (MTT) reagent (Invitrogen, USA). The cells at a density of 8 x $10^3$ cells/well were seeded in 96 well plates. After 24 h, various concentrations of HCFP aqueous (5–1,500 μg/mL) and methanolic (4–18 μg/mL) extracts were added to the cells and incubated for 24 h. The MTT solution was added to each well and incubated for 2 h at 37˚C. After removing the solution, each well was added with DMSO to dissolve the formazan dye. The absorbance of formazan was measured using the microplate reader (Bio-Rad, USA) at 550 nm and 655 nm as a reference wavelength for subtraction of optical density caused by cell debris.

## Nitrite determination

The nitrite concentration in the culture medium of treated and untreated RAW264.7 cells was measured as an indicator of NO production according to Griess reaction [19]. Briefly, the cells ($1x10^5$ cells/well) were seeded into 24-well plates for 24 h, and then pre-treated cells with various concentrations of HCFP aqueous (25–750 μg/mL) and methanolic (4–12 μg/mL) extracts for 2 h. After 2 h incubation, the cells were incubated with LPS (1 μg/mL) for 24 h. The treatment with diclofenac (DCF; 25 μg/mL) was used as a positive control. The cultured medium was then collected and mixed with an equal volume (1:1) of Griess reagent (Invitrogen, USA). After 10 min incubation at room temperature, the absorbance at 550 nm was measured using a microplate reader.

## Prostaglandin E2 ($PGE_2$) determination

The $PGE_2$ metabolite is measured by using an enzyme immunoassay (EIA) kit (Abnova, Taiwan) based on the conversion of all major $PGE_2$ metabolite into a single stable derivative. The cells (1 x $10^5$ cells/well) were seeded into 24-well plates and cultured for 24 h. The cells were pre-treated with various concentrations of HCFP aqueous (25–750 μg/mL) and methanolic (4–12 μg/mL) extracts for 2 h and thereafter incubated with LPS (1 μg/mL) for 24 h. The treatment with diclofenac (DCF; 25 μg/mL) was used as a positive control. Subsequently, $PGE_2$ concentration in a culture medium was determined with $PGE_2$ EIA kit according to the manufacturer's instructions.

## Western blot analysis

RAW264.7 cells (1 x $10^6$ cells) were seeded into a 5.5-cm culture dish and cultured for 24 h. Cells were pre-treated with various concentrations of HCFP aqueous (25–750 μg/mL) and methanolic (4–12 μg/mL) extracts for 2 h and then incubated with LPS (1 μg/mL) for 24 h. The treatment with diclofenac (DCF; 25 μg/mL) was used as a positive control. The treated

cells were harvested and lysed with lysis buffer (25 mM Tris-HCl pH 7.6, 150 mM NaCl, 5mM EDTA, 1% NP-40, 1% sodium deoxycholate, 0.1% SDS) for 1 h on ice. The protein concentration was determined by Bradford protein assay (Bio-Rad, USA). Equal amounts of protein (30 μg) were loaded and separated on 12% SDS-polyacrylamide gel and afterward the proteins were transferred to the PVDF membrane. The membrane was blocked with a blocking solution, 5% skim milk in phosphate-buffered saline containing Tween-20 (PBST), for 1 h at room temperature, and then incubated with monoclonal anti-iNOS, anti-COX-2, anti-β-Actin (1:1000 dilutions, Cell signaling, Germany) for overnight at 4˚C. The blots were washed twice with PBST and then incubated with horseradish peroxidase (HRP)-conjugated secondary antibody (1:1000 dilutions, Cell signaling, Germany) for 2 h at room temperature. Blots were washed again twice with PBST and PBS, respectively. The protein bands were visualized using ECL detection reagent (GE healthcare, UK).

## RNA isolation and RT-PCR analysis

Cells ($1 \times 10^6$ cells) were seeded into a 5.5-cm culture dish and cultured for 24 h. Cells were pre-treated with various concentrations of HCFP aqueous (25–750 μg/mL) and methanolic (4–12 μg/mL) extracts for 2 h and then incubated with LPS (1 μg/mL) for 6 h. Total RNA from treated cells was isolated using Trizol reagent (Invitrogen, USA) according to the manufacturer's protocol and the RNA was kept at -70˚C until used. Total RNA (1 μg) was used for reverse transcription reaction using M-MuLV reverse transcriptase (NEB, UK), 0.5 μM specific reverse primer, deoxyribonucleotide triphosphate (dNTP, 0.2 mM) and 1 U RNase inhibitor. The reaction was incubated at 42˚C for 1 h and the M-MuLV reverse transcriptase was then inactivated by heating at 65˚C for 20 min. The PCR reactions were carried out in a total volume of 25 μl containing 2.5 U of *Taq* DNA polymerases, 0.2 mM dNTP, 1X reaction buffer, and 0.5 μM of forward and reverse primers as listed in S1 Table. After initial denaturation for 30 sec at 95˚C, the amplification by 30 cycles of 94˚C for 45 sec (denaturing), 50–55˚C for 45 sec (annealing), 72˚C for 45 sec (extension), was carried out. The PCR products were analyzed by 1.5% agarose gel electrophoresis. The level of mRNA expression was quantitated by Quantity One software 4.4.1 (Bio-Rad) using β-actin band intensity as the internal control.

## Determination of pro-inflammatory cytokines (IL-1β, IL-6 and TNF-α)

Cells ($1 \times 10^5$ cells/well) were seeded into 24 well plates and cultured for 24 h, and then pre-treated with various concentrations of HCFP aqueous (25–750 μg/ml) and methanolic (4–12 μg/mL) extracts for 2 h. Thereafter, the cells were incubated with LPS (1 μg/mL) for 24 h. The levels of these cytokines in the cultured medium of treated RAW264.7 cells were quantified using ELISA kits according to the manufacturer's instructions. The absorbance at 450 nm was measured using the fluorescence microplate reader (SpectraMax M5, Molecular Devices, USA), and 570 nm was used as a reference wavelength.

## *In vivo* experiment

Animal studies were performed in obligation with the Institutional Animal Care and Use Committee at Khon Kaen University, Khon Kaen, Thailand (*Approval ID*: *AE101/60*) and were performed according to guidelines established by the Ethical Principles and Guidelines for the Use of Animals for scientific purposes, National Research Council of Thailand. All experiments were carried out with six animals in each group. In this study, carrageenan-induced inflammation in the rat paw was used as a model system for *in vivo* anti-inflammatory study. Male Wistar rats (250–300 g) were divided into four different groups, (1) negative control (carrageenan-treated), (2) HCFP 1 (concentration 1-treated), (3) HCFP 2 (concentration 2-treated), and (4) positive control

(Diclofenac-treated). HCFP (3.08 and 6.16 mL/kg) and Diclofenac (150 mg/kg) were administered by orally 1 h before carrageenan induction. The rats received a sub-plantar injection of 100 μL of 1% (w/v) suspension of carrageenan lambda in the right hind paw. The volume of rat paw in all animals was measured at 1, 2 and 3 h after carrageenan injection by using Plethysmometer (Ugo Basile Model 7140, Italy). After λ-carrageenan injection and measurement the volume of rat paw at three hours, the rats were sacrificed by injecting pentobarbital sodium anesthetic. The results were expressed as the changes in paw volume from the baseline value. The percentage of paw edema was calculated using the following equation:

$$\%\text{Paw edema} = (\text{V} - \text{Vi}) \times 100/\text{Vi}$$

Where V = Paw thickness after carrageenan injection and Vi = Paw thickness at 0 time.

## HPLC analysis

Phenolic acid compositions in HCFP aqueous and methanolic extracts were analyzed by using reverse-phase HPLC as previously described [20], with some modifications. The columns used to identify phenolic acids in HCFP aqueous and phenolic extracts were Inertsil®-ODS-4 C18 column (4.6 mm i.d. x 250 mm, 5 μm particle size) and Waters system C18 column (3.9 mm i.d. x 150 mm, 5 μm particle size), respectively, due to availability of the columns at Facilities Service Center, Faculty of Science, Khon Kaen University, Thailand. The linear gradient of solvents A (100% acetonitrile) and B (1% acetic acid in deionized water) for Inertsil®-ODS-4 C18 column was as follows: 0 min, 3% A: 97% B; 5 min, 8% A: 92% B; 15 min, 8% A: 92% B; 25 min, 10% A: 90% B; 55 min, 10% A: 90% B. The linear gradient of solvents for Waters system C18 column was as previously described [20]. The internal standard (*m*-hydroxybenzaldehyde; 1 μg) was used to ensure the accuracy of phenolic acid identification.

## Statistical analysis

Data are expressed as mean ± S.D. form two or three independent experiments. The data analysis was performed by one-way ANOVA with Duncan's post hoc test. Differences were considered to be significant at $p < 0.05$.

# Results

## Effect of HCFP aqueous and methanolic extracts on cell viability in RAW264.7 cells

To study the effect of HCFP aqueous and methanolic extracts on inflammatory responses *in vitro*, RAW264.7 macrophage cells, which play an important role in the maintenance of tissue homeostasis, were used as a model system. The concentrations of both extracts that had no adverse effects on the growth of RAW264.7 cells were determined using MTT assay. Both HCFP aqueous (Fig 1A) and methanolic (Fig 1B) extracts showed no toxicity against RAW264.7 cells at the concentration ranges of 5–750 and 4–12 μg/mL, respectively. The cell viability of more than 90% as compared with a control group was considered non-toxic. Thus, these concentration ranges of HCFP aqueous and methanolic extracts were selected for further study on the anti-inflammatory effect.

## Effect of HCFP aqueous and methanolic extracts on nitric oxide (NO) production of LPS-stimulated RAW264.7 cells

NO is a pro-inflammatory mediator produced by activated macrophages that induce inflammation under pathological conditions [21]. To investigate the effect of HCFP aqueous and

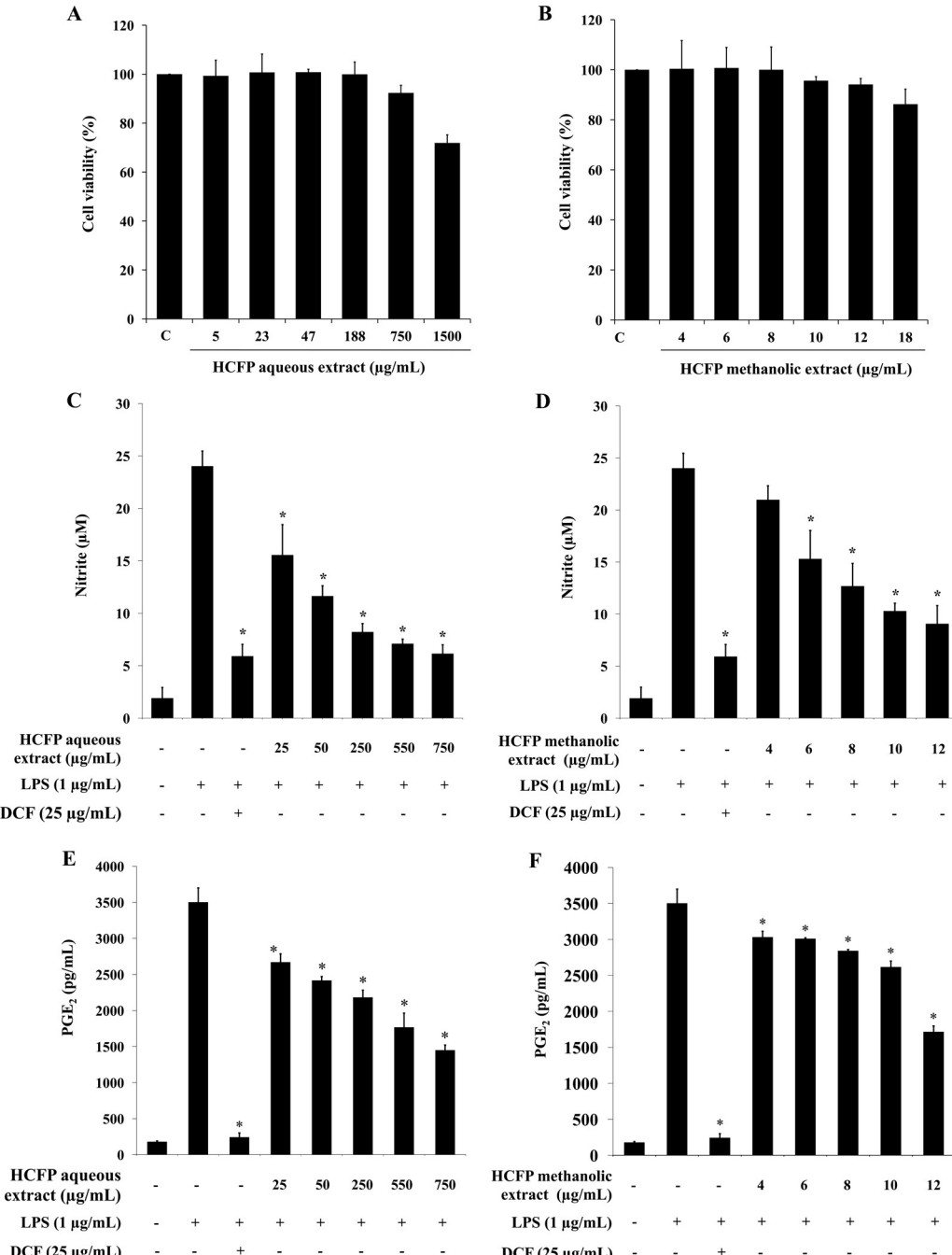

**Fig 1.** Effect of aqueous and methanolic extracts of HCFP on RAW264.7 cell viability (A, B), NO production (C, D) and PGE$_2$ levels (E, F) in LPS-stimulated RAW264.7 cells. RAW264.7 cells were incubated with aqueous (5–1,500 μg/mL) and methanolic (4–18 μg/mL) extracts for 24 h. Cell viability was assessed by MTT assay. The results were reported as a percentage of cell viability compared with untreated controls and expressed as mean ± S.D. of three independent experiments. For determinations of NO production and PGE$_2$ levels in LPS-stimulated RAW264.7 cells, the cells were pre-treated with indicated concentrations of aqueous (C, E) and methanolic (D, F) extracts for 2 h and then stimulated with LPS (1 μg/mL) for 24 h. The nitrite production and PGE$_2$ levels in cultured medium were determined by using Griess reagent and PGE$_2$ EIA kit, respectively. Statistically significant inhibition of NO production and reduction of PGE$_2$ levels (*p < 0.05) were found as compared with the LPS group. Data were obtained from three and two independent experiments, respectively.

methanolic extracts on NO production, RAW264.7 cells were pre-treated with aqueous (25–750 μg/mL) and methanolic (4–12 μg/mL) extracts and thereafter stimulated with LPS (1 μg/mL). NO production was determined by the measurement of nitrite released into the cultured medium using the Griess reagent. The NSAID drug diclofenac (DCF; 25 μg/mL), a positive control for comparing the activity of HCFP extracts, inhibited NO release by 75.40% in LPS-stimulated RAW264.7 macrophages (Fig 1C and 1D). The maximum (750 μg/mL) and minimum (25 μg/mL) concentrations of the HCFP aqueous extract reduced NO production by 74.40% and 35.22%, respectively (Fig 1C). Notably, the HCFP methanolic extract at maximum (12 μg/mL) and minimum (4 μg/mL) concentrations reduced NO production by 62.26% and 12.66%, respectively (Fig 1D). Accordingly, our results showed that both HCFP aqueous and methanolic extracts inhibited NO production in a concentration-dependent manner in LPS-stimulated RAW264.7 cells.

## Effect of HCFP aqueous and methanolic extracts on $PGE_2$ production in LPS-stimulated RAW264.7 cells

$PGE_2$ produced from arachidonic acid through the function of cyclooxygenase (COX) enzymes during inflammatory responses exacerbates the inflammatory process through several signaling modules [22]. We sought to investigate the inhibitory effect of HCFP extracts on $PGE_2$ levels in LPS-stimulated macrophages, which may be an effective strategy for treating inflammatory disorders. Similar to the effect on NO production, both HCFP aqueous and methanolic extracts caused a dose-dependent inhibition of $PGE_2$ production in LPS-stimulated RAW264.7 cells (Fig 1E and 1F). $PGE_2$ level was increased to 3,503.11 pg/mL in LPS treatment, whereas in the absence of LPS, $PGE_2$ level was reduced to 179.42 pg/mL. $PGE_2$ levels were significantly reduced in the cells treated with aqueous (25–750 μg/mL) (Fig 1E) and methanolic (4–12 μg/mL) (Fig 1F) extracts, especially at the highest concentrations tested (58.59% and 51.00% reduction by aqueous (750 μg/mL) and methanolic (12 μg/mL) extracts, respectively). However, diclofenac (25 μg/mL) inhibited $PGE_2$ production by 93.04% in LPS-stimulated RAW264.7 macrophages (Fig 1E and 1F).

## Effect of HCFP aqueous and methanolic extracts on expressions of inducible nitric oxide synthase (iNOS) and cyclooxygenase-2 (COX-2) at both mRNA and protein expression levels in LPS-stimulated RAW264.7 cells

NO is produced from the conversion of L-arginine to L-citrulline by iNOS [23], whereas $PGE_2$ production is mediated by COX-2 [22]. We sought to investigate whether the observed inhibition of HCFP aqueous and methanolic extracts on LPS-induced NO and $PGE_2$ production (Fig 1C–1F) was related to the modulation of iNOS and COX-2 using RT-PCR and Western blot analysis. The mRNA and protein expression levels of iNOS and COX-2 were minimally detected and undetectable, respectively, whereas their levels were significantly increased by LPS treatments (Figs 2 and 3, respectively). iNOS mRNA induction was significantly suppressed by pre-incubation with the NSAID drug diclofenac (25 μg/mL), however, the greater suppression was observed for pre-incubation with both HCFP aqueous (25–750 μg/mL) (Fig 2A and 2B) and methanolic (6–12 μg/mL) extracts (Fig 2C and 2D). iNOS protein level was significantly decreased for pre-incubation with both HCFP aqueous (25–750 μg/mL) (Fig 3A and 3C) and methanolic (4–12 μg/mL) (Fig 3B and 3D) extracts. Pre-incubation with diclofenac did not cause a significant decrease in either mRNA (Fig 2) or protein (Fig 3) levels of COX-2. Similarly, COX-2 mRNA induction was not significantly suppressed by pre-incubation with both HCFP aqueous (50–750 μg/mL) (Fig 2A and 2B) and methanolic (8–12 μg/mL)

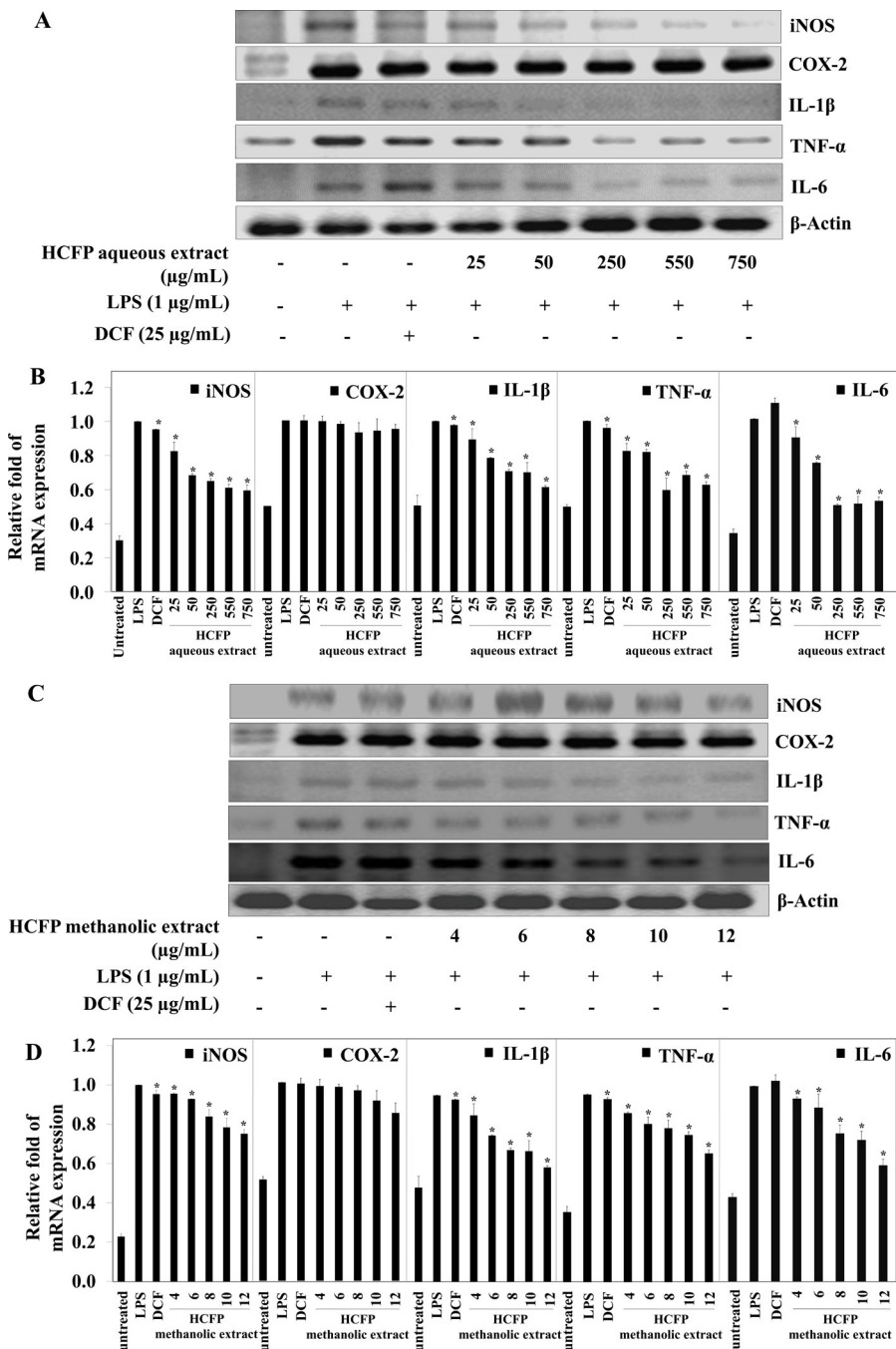

**Fig 2. Effect of aqueous and methanolic extracts of HCFP on mRNA expression of iNOS, COX-2, IL-1β, TNF-α, and IL-6 in LPS-stimulated RAW264.7 cells.** The cells were pre-treated with indicated concentrations of aqueous and methanolic extracts for 2 h and then stimulated with LPS (1 μg/mL) for 6 h. The mRNA expression levels of aqueous extract-treated (A) and methanolic extract-treated (C) cells were determined by reverse transcription-PCR. Bar graphs showed the relative fold of mRNA expression of aqueous extract-treated (B) and methanolic extract-treated (D) cells, *p < 0.05 compared with the LPS group.

(Fig 2C and 2D) extracts. COX-2 protein level was not significantly decreased for pre-incubation with both HCFP aqueous (50–750 μg/mL) (Fig 3A and 3C) and methanolic (4–12 μg/mL) (Fig 3B and 3D) extracts. The above results suggest that inhibition of NO and PGE₂ production

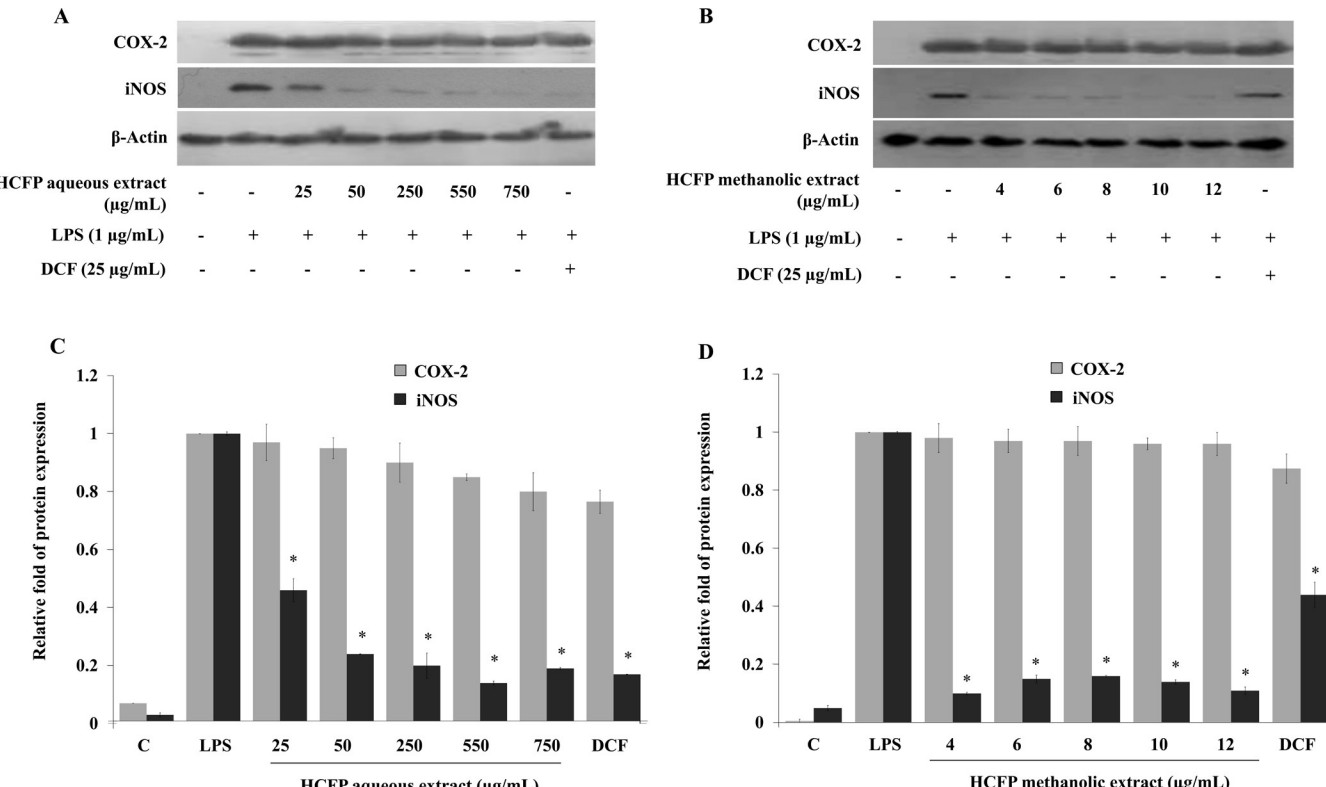

**Fig 3.** Effect of aqueous (A) and methanolic (B) extracts of HCFP on protein levels of iNOS and COX-2 in LPS-stimulated RAW264.7 cells. The cells were pre-treated with the indicated concentration of aqueous extract or phenolic extract for 2 h and then stimulated with LPS (1 µg/mL) for 24 h. The protein expression levels were analyzed by western blot. Bar graphs showed the relative fold of protein expression in aqueous- (C) and methanolic- (D) treated cells. Data were expressed as mean ± S.D. *p < 0.05 compared with the LPS group.

by both HCFP aqueous and methanolic extracts is related to the down-regulation of iNOS but not COX-2.

### Effect of HCFP aqueous and methanolic extracts on expressions of IL-1β, TNF-α and IL-6 at both mRNA and protein expression levels in LPS-stimulated RAW264.7 cells

Interaction between LPS and the membrane receptor CD14 of macrophages caused the induction of pro-inflammatory cytokines including IL-1β, TNF-α and IL-6 [24]. These pro-inflammatory cytokines have been considered as targets for anti-inflammatory therapies [25]. To investigate the anti-inflammatory action of aqueous and methanolic extracts of *H. cordata* fermentation product, the production of the pro-inflammatory cytokines was evaluated by both RT-PCR and ELISA. The mRNA levels of IL-1β, TNF-α, and IL-6 were up-regulated in LPS treated cells compared with untreated controls (Fig 2A–2D). Diclofenac treatment caused a significant decrease in mRNA levels of IL-1β and TNF-α but not IL-6 (Fig 2A–2D), whereas treatments with both HCFP aqueous (25–750 µg/mL) (Fig 2A and 2B) and methanolic (4–12 µg/mL) (Fig 2C and 2D) extracts caused a significant decrease in IL-1β, TNF-α, and IL-6 mRNA levels. Based on ELISA results, induction of IL-1β, TNF-α, and IL-6 protein levels was dose-dependently suppressed by pre-incubation with both HCFP aqueous (25–750 µg/mL) (Fig 4A, 4C and 4E) and methanolic (4–12 µg/mL) (Fig 4B, 4D and 4F) extracts. LPS-induced IL-1β production was inhibited by 68.78 or 62.09% when treated with the highest

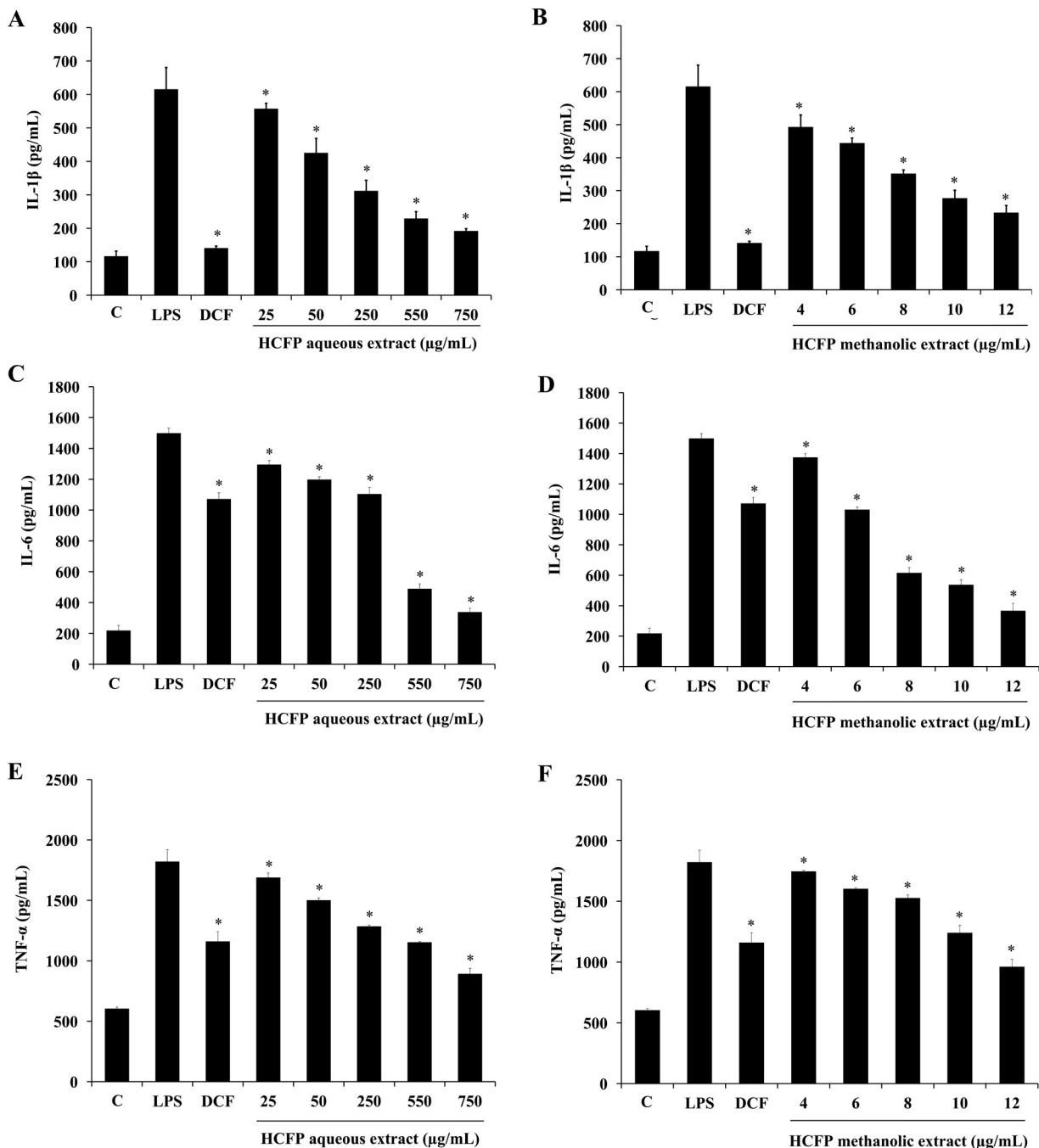

**Fig 4.** Effect of HCFP on production of IL-1β (A, B), IL-6 (C, D) and TNF-α (E, F) in LPS-stimulated RAW264.7 cells. The cells were pre-treated with the indicated concentration of aqueous extract or phenolic extract for 2 h then stimulated with LPS (1 μg/mL) for 24 h. The IL-1β, IL-6, and TNF-α in cultured medium were determined by ELISA kits. Data were expressed as the mean±S.D. of two independent experiments. Statistically significant inhibitions of IL-1β, IL-6 and TNF-α production (*p < 0.05) were found as compared with the LPS group.

concentration of HCFP aqueous (750 μg/mL) (Fig 4A) or methanolic (12 μg/mL) (Fig 4B) extract, respectively. LPS-induced IL-6 production was inhibited by 77.38 or 75.53% when treated with the highest concentration of HCFP aqueous (Fig 4C) or methanolic (Fig 4D) extract, respectively. In addition, LPS-induced TNF-α production was inhibited by 50.99 or

47.21% when treated with the highest concentration of HCFP aqueous (Fig 4E) or methanolic (Fig 4F) extract, respectively.

### Effect of HCFP on carrageenan-induced paw edema in Wistar rats

To evaluate the *in vivo* anti-inflammatory activity of the HCFP Dokudami, the carrageenan-induced paw edema model was chosen as it is sensitive and reproducible *in vivo* test for NSAID drugs and has long been established as a valid model for studying new anti-inflammatory drugs [26]. The formation of paw edema was gradually increased within the first hour after carrageenan injection (Fig 5). A common clinical NSAID drug diclofenac (DCF) was used as a positive control pre-treated at 150 mg/kg. DCF significantly ($p < 0.05$) reduced paw edema after 2 h carrageenan stimulation. Similarly, the HCFP (3.08 and 6.16 mL/kg) also significantly ($p < 0.05$) reduced paw edema after 2 h carrageenan stimulation. However, pre-treatment of HCFP at a concentration of 6.16 mL/kg caused a significant ($p < 0.05$) reduction of paw edema after 1 h carrageenan stimulation.

### Quantification of phenolic composition in HCFP by HPLC

The component profiles of HCFP aqueous and methanolic extracts were analyzed by HPLC. The representative chromatograms were shown in Fig 6. Six phenolic acids were identified in HCFP aqueous extract including *p*-hydroxybenzoic, vanillic, syringic, *p*-coumaric, ferulic and gallic acids (Fig 6B and Table 1). Whereas, seven phenolic acids were identified in methanolic extract of HCFP including protocatechuic, *p*-hydroxybenzoic, vanillic, syringic, *p*-coumaric, ferulic and sinapinic acids (Fig 6D and Table 1). Among the identified phenolic acids of HCFP, the most abundant phenolic acid in both aqueous and methanolic extracts was syringic acid (Table 1).

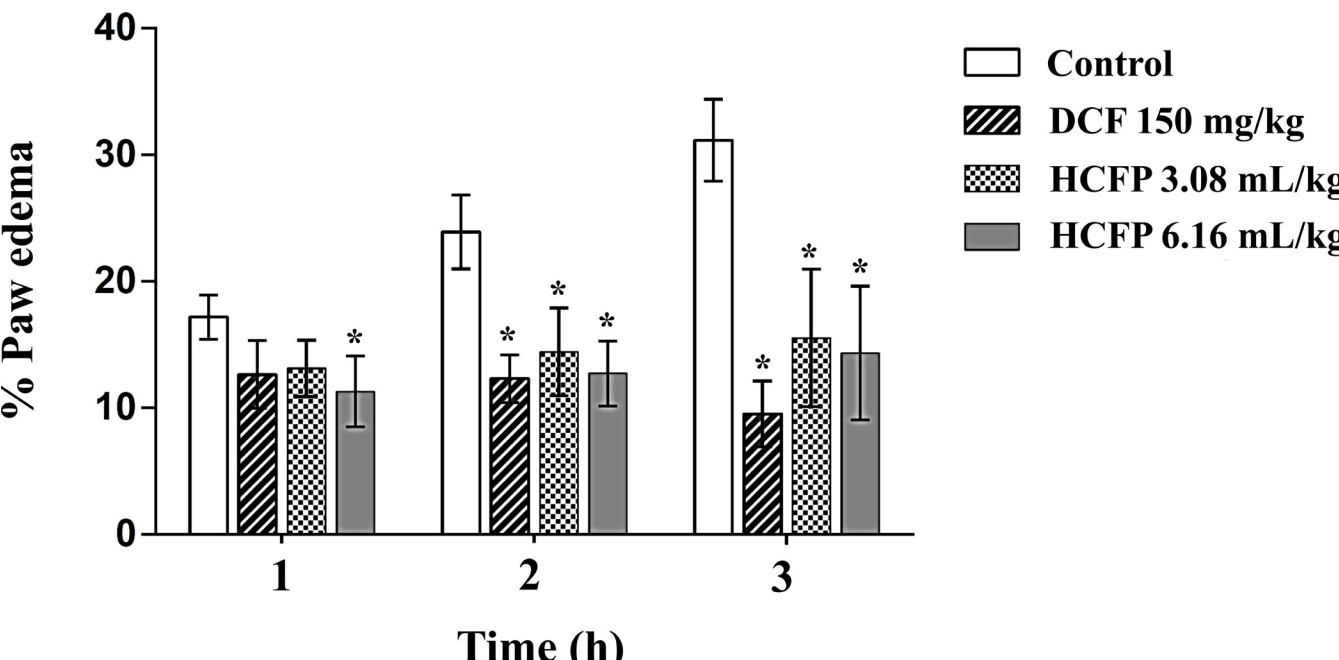

**Fig 5. Effect of HCFP and DCF on carrageenan-induced paw edema in Wistar rats.** Bar graphs show percentages of changes in paw edema. Data are expressed as mean ± SD of n = 6 rats/group. Asterisk "*" indicates a significant difference at $p < 0.05$ as compared with the control group.

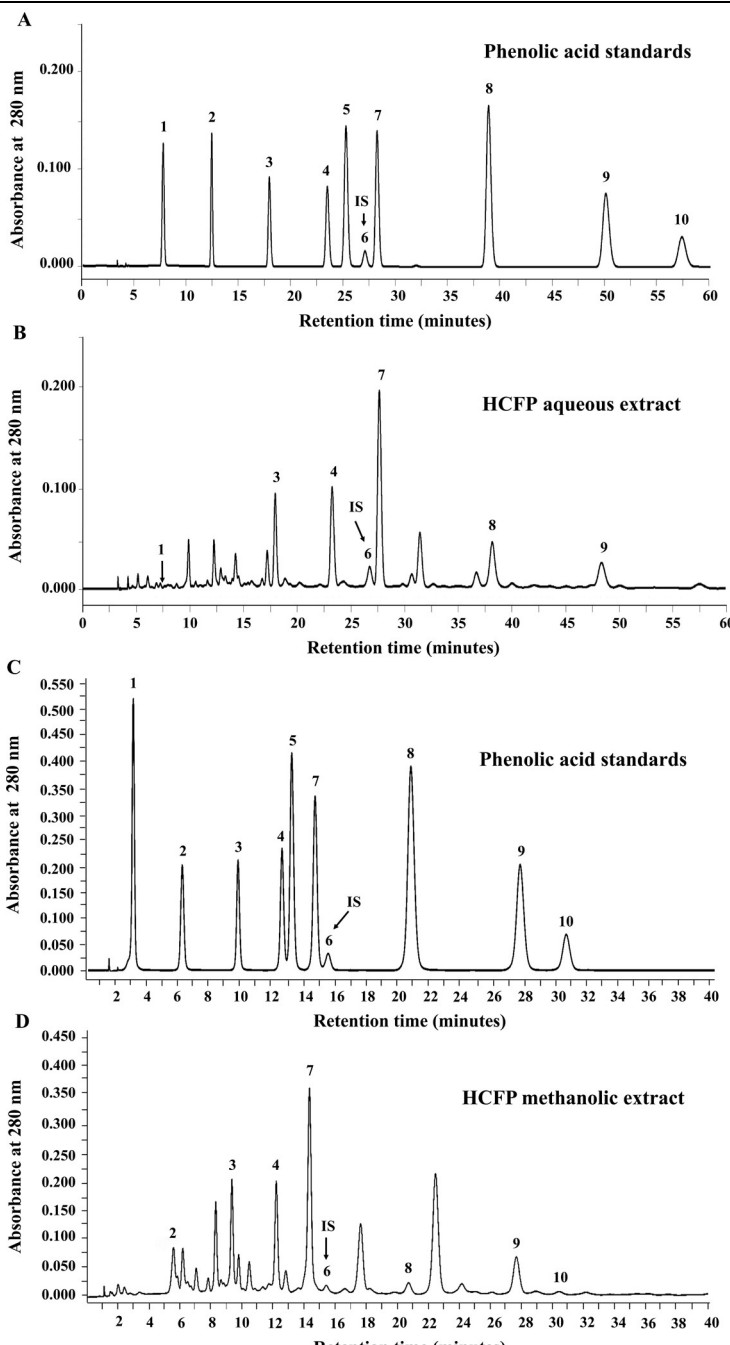

**Fig 6.** HPLC chromatograms of phenolic acid standards (A, C) and base hydrolyzed HCFP aqueous (B) and methanolic (D) extracts, where 1 = gallic acid, 2 = protocatechuic acid, 3 = *p*-hydroxybenzoic acid, 4 = vanillic acid, 5 = caffeic acid, 6 = *m*-hydroxybenzaldehyde, 7 = syringic acid, 8 = *p*-coumaric acid, 9 = ferulic acid and 10 = sinapinic acid. The *m*-hydroxybenzaldehyde was used as an internal standard (I.S.).

## Discussion

This study was based on the extensive use of industrial HCFPs as a dietary supplement in Thailand without scientific testing on their biological properties. Therefore, we aimed at investigating the anti-inflammatory activity of a commercialized fermented broth of *H. cordata* both *in vitro* (LPS-induced RAW264.7 model) and *in vivo* (carrageenan-induced paw edema model).

**Table 1. Phenolic acid compositions of aqueous and methanolic extracts of HCFP.**

| Samples | Phenolic acids[a] ($\mu g/g$ of HCFP) | | | | | | | |
|---|---|---|---|---|---|---|---|---|
| | Gallic acid | Protocatechuic acid | p-Hydroxybenzoic acid | Vanillic acid | Syringic acid | p-Coumaric acid | Ferulic acid | Sinapinic acid |
| HCFP aqueous extract | 14.36±4.96 | n.d. | 64.30±5.68 | 79.97±11.30 | 88.23±12.83 | 20.53±0.94 | 32.22±4.94 | n.d. |
| HCFP methanolic extract | n.d. | 312.54±0.08 | 774.21±0.12 | 869.95±0.91 | 2268.34±0.44 | 162.87±0.07 | 670.35±0.14 | 66.10±0.01 |

[a]Results are expressed as means ± SD of three determinations.

n.d., not detected.

To study the anti-inflammatory potential of the industrial HCFP, the inherent cytotoxic effects of the HCFP extracts on the cellular model used in this study were predetermined using MTT assay. All concentrations of the HCFP extracts used in this study induced negligible cytotoxic effects on RAW264.7 macrophages (cell viability > 90%) (Fig 1A and 1B), indicating that the inhibitory effect of both extracts on production of inflammatory mediators is not attributed to cytotoxicity. LPS from gram-negative bacteria has been shown to possess a dose-dependent cytotoxic activity in RAW264.7 cells [13], therefore, the non-toxic concentration of LPS was predetermined (Data not shown). LPS at concentration of 1 $\mu g/ml$ was used in the present study, and not toxic to RAW264.7 cells, consistent with the result from previous study [27].

In the present study, we reported for the first time that the industrial HCFP possessed anti-inflammatory activity. Both HCFP aqueous and methanolic extracts successfully inhibited the production of inflammatory mediators (NO and $PGE_2$). During the inflammatory process, large amounts of the pro-inflammatory mediators like nitric oxide and prostaglandins $E_2$ are generated by the inducible nitric oxide synthase (iNOS) and cyclooxygenase (COX-2), respectively. Nitric oxide is one of pro-inflammatory mediator that responds to pathogenic infections. During the inflammatory process, NO is generated by macrophages to eliminate foreign pathogens, recruiting other cells to the infected area and subsequently resolving the inflammation. However, the excessive amount of NO is also harmful to normal tissue surrounding the infected area because it binds with other superoxides radical and acts as a reactive radical to damage normal cell function. Gram negative bacterial LPS is well known to increase iNOS expression and NO production, leading to the initiation of an inflammatory response [28]. Therefore, the inhibition of NO production is a key therapeutic consideration in both searching for anti-inflammatory agents and developing a novel treatment for inflammatory disorders. In the present study, both aqueous and methanolic extracts of *H. cordata* fermentation product reduced NO production in LPS-stimulated RAW264.7 cells in a concentration dependent manner (Fig 1C and 1D). The decreased NO production was correlated well with the dose-dependent decrease of iNOS mRNA (Fig 2) and protein (Fig 3) levels.

$PGE_2$, an inflammatory mediator, is produced by the metabolism of arachidonic acid by COX enzymes at inflammatory sites [29]. $PGE_2$ production increased following LPS treatment. After the RAW264.7 cells were pre-treated with HCFP aqueous and methanolic extracts, the $PGE_2$ levels in LPS-stimulated RAW264.7 cells were decreased in a dose-dependent fashion (Fig 1E and 1F). However, the NSAID diclofenac (25 $\mu g/mL$) exhibited more potent inhibitory activity against $PGE_2$ production than HCFP extracts at all concentrations tested. The decreased $PGE_2$ production (Fig 1E and 1F) in both DCF and HCFP treatments was not correlated well with the COX-2 mRNA (Fig 2) and protein (Fig 3) levels. The reduced $PGE_2$ levels may be due to the inhibition of COX-2 activity by DCF and HCFP treatments. Indeed, DCF has been shown to selectively inhibit COX-2 activity [30].

TNF-α, IL-1β, and IL-6 are the main pro-inflammatory cytokines that are primarily produced by macrophages and have various pro-inflammatory effects on many cell types [31,32]. Over-production of TNF-α caused the release of various inflammatory mediators including NO, PGE$_2$, IL-1β and IL-6. Excessive production of cytokines (TNF-α, IL-1β and IL-6) has linked in several physiological effects, including septic shock, inflammation and cytotoxicity [33]. Thus, the inhibition of cytokine production or function is a key mechanism in the control of inflammation [34]. In the present study, aqueous and methanolic extracts of *H. cordata* fermentation product reduced production of cytokines in LPS-stimulated RAW264.7 cells in a concentration-dependent manner both at mRNA (Fig 2) and protein (Fig 4) levels.

Carrageenan-induced paw edema is an animal model suitable for evaluating inhibition of edema. Biphasic edema induced by carrageenan [35], includes the first phase (1 h) involving the release of serotonin and histamine and the second phase (over 1 h) mediating by prostaglandins, cyclooxygenase products. In the present study, both doses of the industrial HCFP significantly reduced paw edema at 2 and 3 h after carrageenan injection. This finding suggests that HCFP produces an anti-edematous effect during the second phase which is similar to DCF (Fig 5). Interestingly, the highest dose of HCFP (6.16 mL/kg) showed a significant reduction of paw edema at 1 h, suggesting an anti-edematous effect during the first phase. Further animal study on the mechanism underlying inhibition of the first/second phase edema is of interest.

The identification of active components in HCFP extracts is an important pharmacological goal. Our HPLC results demonstrated that the amount of all identified phenolic acids in HCFP methanolic extract were much greater than those in HCFP aqueous extract (Table 1). Syringic acid was present in the greatest amounts in both HCFP aqueous (88.23 μg/g of extract) and methanolic (2,268.34 μg/g of extract) extracts, followed by vanillic, *p*-hydroxybenzoic, and ferulic acids, respectively (Fig 6 and Table 1). Yoo et al. [36] demonstrated that syringic, vanillic, *p*-hydroxybenzoic and ferulic acids. In addition, *p*-coumaric acid has been shown to possess anti-inflammatory activity both *in vitro* and *in vivo* [37, 38]. Accordingly, these phenolic acids may contribute to HCFP-mediated inhibition of the production of inflammatory cytokines and mediators. Phenolic acids in the water-soluble constituents of *H. cordata* fermentation product were previously identified and quantified [39], but their amounts were greater than those found in the present HCFP aqueous extract (Table 1). The discrepancy may be due to a lot-to-lot variation of the industrial HCFP. Further study on synergistic anti-inflammatory effects of HCFP individual phenolic acids both *in vitro* and *in vivo* is of interest.

## Conclusions

Our results demonstrated that the aqueous and methanolic extracts of *H. cordata* fermentation product possessed anti-inflammatory activity by inhibiting the production of NO, PGE$_2$ and inflammatory cytokines (TNF-α, IL-1β, IL-6) in LPS-stimulated RAW264.7 cells. The anti-inflammatory activity of the industrial HCFP was confirmed by the inhibition of inflammation in the carrageenan-induced rat paw edema model. Our results suggest that this industrial HCFP may be considered as an anti-inflammatory dietary supplement. The health benefits of this industrial HCFP warrant further clinical studies.

## Supporting information

**S1 Table. Primer list.**
(DOCX)

**S1 Raw images.**
(PDF)

## Acknowledgments

We thank Associate Professor Dr. Albert J. Ketterman for proofreading the manuscript. We are also thankful to Dr. Prasan Swatsitang for providing some phenolic acid standards.

## Author Contributions

**Conceptualization:** Thanaset Senawong.

**Data curation:** Gulsiri Senawong.

**Funding acquisition:** Thanaset Senawong.

**Investigation:** Khanutsanan Woranam, Suppawit Utaiwat.

**Methodology:** Khanutsanan Woranam.

**Project administration:** Thanaset Senawong.

**Resources:** Gulsiri Senawong, Thanaset Senawong.

**Supervision:** Gulsiri Senawong, Sirinda Yunchalard, Jintana Sattayasai.

**Validation:** Thanaset Senawong.

**Writing – original draft:** Khanutsanan Woranam, Gulsiri Senawong, Thanaset Senawong.

**Writing – review & editing:** Gulsiri Senawong, Sirinda Yunchalard, Jintana Sattayasai, Thanaset Senawong.

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
