## [Decision Letter · Decision Letter 0]

17 Dec 2019

PONE-D-19-31936

Anti-inflammatory activity of the dietary supplement Houttuynia cordatafermentation product in RAW264.7 cells and Wistar rats

PLOS ONE

Dear Dr. Senawong,

Thank you for submitting your manuscript to PLOS ONE. After careful consideration, we feel that it has merit but does not fully meet PLOS ONE’s publication criteria as it currently stands. Therefore, we invite you to submit a revised version of the manuscript that addresses the points raised during the review process.

We would appreciate receiving your revised manuscript by Jan 31 2020 11:59PM. To enhance the reproducibility of your results, we recommend that if applicable you deposit your laboratory protocols in protocols.io, where a protocol can be assigned its own identifier (DOI) such that it can be cited independently in the future. For instructions see: http://journals.plos.org/plosone/s/submission-guidelines#loc-laboratory-protocols

We look forward to receiving your revised manuscript.

Kind regards,

Walid Elfalleh, Ph.D

Academic Editor

PLOS ONE

Journal Requirements:

2) At this time, we request that you  please report additional details in your Methods section regarding animal care, as per our editorial guidelines:

a) Please provide details of animal welfare (e.g., shelter, food, water, environmental enrichment)

b) please describe any steps taken to minimize animal suffering and distress, such as by administering analgesics,

c) please include the method of sacrifice and

d) Please describe the frequency of monitoring and the criteria used to assess animal health and well-being.

3) We also ask that you please provide in your Methods section, additional details regarding the source of the Houttuynia cordata fermentation product used in this study. Please provide the product number, lot number, and full ingredient list to ensure reproducibility of the analyses.

4) Please also clarify whether your animal ethics committee specifically approved this study.

5)  PLOS ONE now requires that authors provide the original uncropped and unadjusted images underlying all blot or gel results reported in a submission’s figures or Supporting Information files. This policy and the journal’s other requirements for blot/gel reporting and figure preparation are described in detail at https://journals.plos.org/plosone/s/figures#loc-blot-and-gel-reporting-requirements and https://journals.plos.org/plosone/s/figures#loc-preparing-figures-from-image-files. When you submit your revised manuscript, please ensure that your figures adhere fully to these guidelines and provide the original underlying images for all blot or gel data reported in your submission. See the following link for instructions on providing the original image data: https://journals.plos.org/plosone/s/figures#loc-original-images-for-blots-and-gels.

6)  Thank you for stating the following financial disclosure:

 [This study was supported by the Research and Researcher for Industry (RRi) project, Thailand Research Fund (TRF), which was cooperated with the Prolac (Thailand) Co., Ltd., Lamphun Province, Thailand.].               

7) Thank you for stating the following in the Competing Interests section:

[The authors have declared that no competing interests exist.].

We note that you received funding from a commercial source: Prolac (Thailand) Co., Ltd

Reviewers' comments:

Reviewer's Responses to Questions

**Comments to the Author**

1. Is the manuscript technically sound, and do the data support the conclusions?

Reviewer #1: Partly

Reviewer #2: Yes

Reviewer #3: Yes

Reviewer #4: Yes

2. Has the statistical analysis been performed appropriately and rigorously? 

Reviewer #1: Yes

Reviewer #2: Yes

Reviewer #3: Yes

Reviewer #4: Yes

3. Have the authors made all data underlying the findings in their manuscript fully available?

Reviewer #1: No

Reviewer #2: Yes

Reviewer #3: Yes

Reviewer #4: Yes

4. Is the manuscript presented in an intelligible fashion and written in standard English?

Reviewer #1: No

Reviewer #2: Yes

Reviewer #3: Yes

Reviewer #4: Yes

5. Review Comments to the Author

Reviewer #1: Title: “Anti-inflammatory activity of the dietary supplement Houttuynia cordatafermentation product in RAW264.7 cells and Wistar rats”

Corresponding Author: Thanaset Senawong

In the paper entitled “Anti-inflammatory activity of the dietary supplement Houttuynia cordatafermentation product in RAW264.7 cells and Wistar rats” by Khanutsanan Woranam et al ., had showed Houttuynia cordatafermentation acts as an Anti-inflammatory RAW264.7 cells and Wistar rats . Overall, the results of this study are of some values in revealing its role in Anti-inflammatory activity.

Below are some examples of issues in which the authors might want to work on.

Many grammar, format and language mistakes are present throughout the manuscript. Manuscript need to have a review by English editor.

Material and method:

Nitrite determination was performed by spectophotmetry or any other method if so could you please explain this why spectrofluorimetric was not used as this method is considered as attractive due to its facility availability, high sensitivity and selectivity, low limits of detection and low-cost.

Primer sequences placed in method section should be added as supplementary table.

Was time course study performed when incubated with LPS (1 μg/mL) in case of RNA isolation and protein extraction.

Result section:

How many sets of experiments were performed for the cell viability assay? Could you add cell pictures before and after treatment?

In figure 2C the expression level of IL-1β when treated HCFP methanolic extract

At 12 (μg/mL) showed decreased level of expression while in quantitativeresults figure 2D showed increased level. Could you please explain which method was used for the quantitative analysis?

Discussion

The discussion part is poorly described it need to be revised and latest references should be added.

The manuscript is not acceptable in the present format. It needs to be extensively revised.

Reviewer #2: The manuscript is providing some nice conclusions about a potent dietary supplement, however, I have a concern about the conclusion. This needs to be rewritten according to the results obtained and should't be so general. Further not a single reference is quoted from the literature published in 2019. Hence, the latest references should be added.

Reviewer #3: The paper is very interesting and designed diliginty in a well professional manner. The paper entitled"Anti-inflammatory activity of the dietary supplement Houttuynia cordatafermentation product in RAW264.7 cells and Wistar rats" is considered a new approach in combining effect of Houttuynia cordatafermentation product both invivo and invitro by carrying out well designed experiments and performing cell viability test by explanation in-details and the authors reported all the kits sources and also the approval number of animal ethics in their university, the statistical analysis was performed professionally , the discussion part is brilliant in explaining the causes of inflammation and the efficacy of the used extract .

So i recommend publication of this paper which deserve publication and will be of great benefit for readers.

Reviewer #4: Thank you to give me the opportunity to review this paper

I found this is interesting.

My comments:

1. I am warried that the HCFP has a degraded the protein or has cytotoxicity on the cells.

2. It suggested to do another viability assay to confirm the result

Thanks

6. PLOS authors have the option to publish the peer review history of their article (what does this mean?). If published, this will include your full peer review and any attached files.

Reviewer #1: No

Reviewer #2: No

Reviewer #3: Yes: Dr. Reham Zakaria Hamza

Reviewer #4: Yes: Abdullah M Alkahtani

---

## [Author Response · Author response to Decision Letter 0]

11 Feb 2020

The response to the referees is as following:

Journal Requirements:

Response: We have checked to ensure that our manuscript meets PLOS ONE's style requirements.

2) At this time, we request that you please report additional details in your Methods section regarding animal care, as per our editorial guidelines:

a) Please provide details of animal (e.g., shelter, food, water, environmental enrichment)

Response: We have provided details of animal regarding shelter, food, water, environmental enrichment in the Cell Culture and Animals section of Materials and Methods (pages 6-7, lines 128-137).

b) please describe any steps taken to minimize animal suffering and distress, such as by administering analgesics,

Response: The researchers have trained for work with animal before doing the research. During conducting the experiment, the researcher worked with animal gently and quickly to reduce pain or suffering of animal. In this study, we did not use the analgesics for animal as we worked with animals in USDA class C pain level. The animal may suffer for a while with no need to administer analgesics.

c) please include the method of sacrifice and

Response: We have added the method of sacrifice in the in vivo experiment section of Materials and Methods (page 12, lines 250-252).

d) Please describe the frequency of monitoring and the criteria used to assess animal health and well-being.

Response: The animal health was checked and monitored in detail by the veterinarian of the Northeast Laboratory Center, Khon Kaen University, every day. The criteria used to assess animal health and well-being included appearance of eyes, noses, mouth, anus, and social activity.

3) We also ask that you please provide in your Methods section, additional details regarding the source of the Houttuynia cordata fermentation product used in this study. Please provide the product number, lot number, and full ingredient list to ensure reproducibility of the analyses.

Response: We have added additional details as …“ The information on plant ingredient and serving suggestion of the HCFP were obtained from the label on its container. The major ingredients of this HCFP are composed of 99.3% (w/w) aerial parts of H. cordata and 0.7 % (w/w) sugar cane powder. Serving suggestion is as follows: 5-15 ml twice a day in the morning before bedtime and before meal. H. cordata was cultivated by the Prolac (Thailand) Co., Ltd. in an organic farm in Chai Badan district, Lopburi province, Thailand. The fermentation product Lot no. 14/5/2015 was used throughout the study.”….in Materials section of Materials and Methods (pages 5-6, lines 107-113).

4) Please also clarify whether your animal ethics committee specifically approved this study.

Response: We have mentioned that the approval number was IACUC-KKU-101/60 in Cell Culture and Animals section of Materials and Methods (page 7, line 137). Here is the approval certificate from animal ethics committee who specifically approved this study.

5) PLOS ONE now requires that authors provide the original uncropped and unadjusted images underlying all blot or gel results reported in a submission’s figures or Supporting Information files. This policy and the journal’s other requirements for blot/gel reporting and figure preparation are described in detail at https://journals.plos.org/plosone/s/figures#loc-blot-and-gel-reporting-requirements and https://journals.plos.org/plosone/s/figures#loc-preparing-figures-from-image-files. When you submit your revised manuscript, please ensure that your figures adhere fully to these guidelines and provide the original underlying images for all blot or gel data reported in your submission. See the following link for instructions on providing the original image data: https://journals.plos.org/plosone/s/figures#loc-original-images-for-blots-and-gels.

Response: The original underlying images are in Supporting Information as “S1_Raw_Images”. 

6) Thank you for stating the following financial disclosure:

 [This study was supported by the Research and Researcher for Industry (RRi) project, Thailand Research Fund (TRF), which was cooperated with the Prolac (Thailand) Co., Ltd., Lamphun Province, Thailand.].

Please state what role the funders took in the study. If the funders had no role, please state: “The funders had no role in study design, data collection and analysis, decision to publish, or preparation of the manuscript.”

Response: We have added the sentence “The funders had no role in study design, data collection and analysis, decision to publish, or preparation of the manuscript” in our cover letter above and in Acknowledgements section of the revised manuscript (pages 27-28, lines 546-547).

7) Thank you for stating the following in the Competing Interests section:

[The authors have declared that no competing interests exist.].

We note that you received funding from a commercial source: Prolac (Thailand) Co., Ltd

Response: We have included the following amended Competing Interests Statement within our cover letter above.

“With the submission of this manuscript I would like to undertake that all of authors have no relation with the Prolac (Thailand) Co., Ltd., who is a co-funder with Thailand Research Fund (TRF) on the Research and Researcher for Industry (RRi) project. The company provided 60,000 baht in cash and the HCFP samples (product Lot no. 14/5/2015), while the TRF provided a Ph.D. scholarship and major funding source for the project. This does not alter our adherence to PLOS ONE policies on sharing data and materials. The funders had no role in study design, data collection and analysis, decision to publish, or preparation of the manuscript. However, the funders have been informed and agreed on publishing a manuscript, sharing data and materials.” 

Reviewers' comments:

Reviewer's Responses to Questions

Comments to the Author

1. Is the manuscript technically sound, and do the data support the conclusions?

Reviewer #1: Partly

Reviewer #2: Yes

Reviewer #3: Yes

Reviewer #4: Yes

2. Has the statistical analysis been performed appropriately and rigorously?

Reviewer #1: Yes

Reviewer #2: Yes

Reviewer #3: Yes

Reviewer #4: Yes

3. Have the authors made all data underlying the findings in their manuscript fully available?

Reviewer #1: No

Reviewer #2: Yes

Reviewer #3: Yes

Reviewer #4: Yes

4. Is the manuscript presented in an intelligible fashion and written in standard English?

Reviewer #1: No

Reviewer #2: Yes

Reviewer #3: Yes

Reviewer #4: Yes

5. Review Comments to the Author

Reviewer #1: Title: “Anti-inflammatory activity of the dietary supplement Houttuynia cordatafermentation product in RAW264.7 cells and Wistar rats”

Corresponding Author: Thanaset Senawong

In the paper entitled “Anti-inflammatory activity of the dietary supplement Houttuynia cordatafermentation product in RAW264.7 cells and Wistar rats” by Khanutsanan Woranam et al., had showed Houttuynia cordata fermentation acts as an Anti-inflammatory RAW264.7 cells and Wistar rats. Overall, the results of this study are of some values in revealing its role in Anti-inflammatory activity.

Below are some examples of issues in which the authors might want to work on.

Many grammar, format and language mistakes are present throughout the manuscript. Manuscript need to have a review by English editor.

Response: The manuscript has been checked for grammatical errors using Grammarly® program (https://www.grammarly.com/) and has been proofreading by native speaker, Associate Professor Dr. Albert J. Ketterman, Institute of Molecular Biosciences, Mahidol University, Salaya Campus, 73170 Thailand.

. 

Material and method:

Nitrite determination was performed by spectophotmetry or any other method if so could you please explain this why spectrofluorimetric was not used as this method is considered as attractive due to its facility availability, high sensitivity and selectivity, low limits of detection and low-cost.

Response: Nitrite determination was performed by spectrophotometry. This method is sensitive and selective color reaction for the determination of nitrite and is popularly used by many researchers. This method has been successfully applied to the determination of trace amounts of nitrite and nitrate in water, soil and pharmaceutical preparations. We have set up this method in our lab due to availability of spectrophotometer instead of fluorospectrometer.

Primer sequences placed in method section should be added as supplementary table.

Response: We have placed the primer sequences to “S1 Table Primer list” in Supporting Information.

Was time course study performed when incubated with LPS (1 μg/mL) in case of RNA isolation and protein extraction.

Response: We have not performed time course study for LPS treatment. We performed 6 h incubation for RNA isolation and 24 h for protein isolation according to our predetermined time course data. 

Result section:

How many sets of experiments were performed for the cell viability assay? Could you add cell pictures before and after treatment?

Response: We performed 3-5 independent experiments for the cell viability assay. The cell morphology was also observed under inverted microscope. Here is an example of cell morphology observed under inverted microscope. However, we cannot not put all images because we have not taken all images during performing the experiments although we have monitored all images.

In figure 2C the expression level of IL-1β when treated HCFP methanolic extract

At 12 (μg/mL) showed decreased level of expression while in quantitative results figure 2D showed increased level. Could you please explain which method was used for the quantitative analysis?

Response: We quantitate the band intensity using Quantity One software 4.4.1 (Bio-Rad). We have checked the result and found that the expression level of IL-1β in figure 2C when treated HCFP methanolic extract at 12 μg/mL showed decreased level of expression, however, the quantitative results in figure 2D did not show an increased level as mentioned by the reviewer 1. The quantitative results in figure 2D actually showed decreased level corresponding to figure 2C. 

Discussion

The discussion part is poorly described it need to be revised and latest references should be added.

Response: We have thoroughly checked the discussion part for a revised manuscript. Two latest references were added as ref. 32 and 38.

The manuscript is not acceptable in the present format. It needs to be extensively revised.

Response: The revised manuscript has been proofreading by native speaker, Associate Professor Dr. Albert J. Ketterman, Institute of Molecular Biosciences, Mahidol University, Salaya Campus, 73170 Thailand.

Reviewer #2: The manuscript is providing some nice conclusions about a potent dietary supplement, however, I have a concern about the conclusion. This needs to be rewritten according to the results obtained and should't be so general. Further not a single reference is quoted from the literature published in 2019. Hence, the latest references should be added.

Response: Thank you for reviewer’s comments. We have added two latest references as ref. 32 and 38.

Reviewer #3: The paper is very interesting and designed diliginty in a well professional manner. The paper entitled"Anti-inflammatory activity of the dietary supplement Houttuynia cordatafermentation product in RAW264.7 cells and Wistar rats" is considered a new approach in combining effect of Houttuynia cordata fermentation product both in vivo and in vitro by carrying out well designed experiments and performing cell viability test by explanation in-details and the authors reported all the kits sources and also the approval number of animal ethics in their university, the statistical analysis was performed professionally , the discussion part is brilliant in explaining the causes of inflammation and the efficacy of the used extract .

So I recommend publication of this paper which deserve publication and will be of great benefit for readers.

Response: Thank you for reviewer’s comments.

Reviewer #4: Thank you to give me the opportunity to review this paper

I found this is interesting.

My comments:

1. I am warried that the HCFP has a degraded the protein or has cytotoxicity on the cells.

Response: We do not think that HCFP has a degraded protein or has cytotoxicity on the cells because we have observed cell morphology under inverted microscope and no cell death was observed at the concentrations tested.

2. It suggested to do another viability assay to confirm the result

Thanks

Response: The cell morphology was also observed under inverted microscope as mentioned above.

---

## [Decision Letter · Decision Letter 1]

5 Mar 2020

Anti-inflammatory activity of the dietary supplement Houttuynia cordata fermentation product in RAW264.7 cells and Wistar rats

PONE-D-19-31936R1

Dear Dr. Senawong,

We are pleased to inform you that your manuscript has been judged scientifically suitable for publication and will be formally accepted for publication once it complies with all outstanding technical requirements.

With kind regards,

Walid Elfalleh, Ph.D

Academic Editor

PLOS ONE

Additional Editor Comments (optional):

Reviewers' comments:

Reviewer's Responses to Questions

**Comments to the Author**

1. If the authors have adequately addressed your comments raised in a previous round of review and you feel that this manuscript is now acceptable for publication, you may indicate that here to bypass the “Comments to the Author” section, enter your conflict of interest statement in the “Confidential to Editor” section, and submit your "Accept" recommendation.

Reviewer #1: All comments have been addressed

Reviewer #4: All comments have been addressed

2. Is the manuscript technically sound, and do the data support the conclusions?

Reviewer #1: Partly

Reviewer #4: Partly

3. Has the statistical analysis been performed appropriately and rigorously? 

Reviewer #1: Yes

Reviewer #4: Yes

4. Have the authors made all data underlying the findings in their manuscript fully available?

Reviewer #1: Yes

Reviewer #4: Yes

5. Is the manuscript presented in an intelligible fashion and written in standard English?

Reviewer #1: Yes

Reviewer #4: Yes

6. Review Comments to the Author

Reviewer #1: thank you for the answering the comments. In my opinion this manuscript is fulfilling the requirements technically for the publication in PLOSone. My comments are accept this manuscript publication.

Reviewer #4: I think this work is nice and the data is publishable and make sure about the most important massages of this paper

7. PLOS authors have the option to publish the peer review history of their article (what does this mean?). If published, this will include your full peer review and any attached files.

Reviewer #1: No

Reviewer #4: Yes: Abdullah M Alkahtnai

---

## [Editor Report · Acceptance letter]

9 Mar 2020

PONE-D-19-31936R1 

Anti-inflammatory activity of the dietary supplement *Houttuynia cordata* fermentation product in RAW264.7 cells and Wistar rats 

Dear Dr. Senawong:

I am pleased to inform you that your manuscript has been deemed suitable for publication in PLOS ONE. Congratulations! Your manuscript is now with our production department. 

With kind regards,

on behalf of

Professor Walid Elfalleh 

Academic Editor

PLOS ONE